# Correlation among Knee Muscle Strength and Self-Reported Outcomes Score, Anterior Tibial Displacement, and Time Post-Injury in Non-Coper Anterior Cruciate Ligament Deficient Patients: A Cross-Sectional Study

**DOI:** 10.3390/ijerph182413303

**Published:** 2021-12-17

**Authors:** Ignacio Manchado, David Alvarez, Luci M. Motta, Gustavo Blanco, Pedro Saavedra, Gerardo L. Garcés

**Affiliations:** 1Hospital Perpetuo Socorro, 35007 Las Palmas, Spain; nachomanchado@gmail.com (I.M.); luci.motta@traumaquir.es (L.M.M.); gustblan@gmail.com (G.B.); 2Department of Ciencias Médicas y Quirúrgicas, University of Las Palmas de Gran Canaria, 35016 Las Palmas, Spain; 3Terapias Acuáticas Canarias, 35011 Las Palmas, Spain; davidalvarez@tacsl.es; 4Mathematics Department, University of Las Palmas de Gran Canaria, 35017 Las Palmas, Spain; pedro.saavedra@ulpgc.es

**Keywords:** anterior cruciate ligament, non-copers, hamstring/quadriceps ratio, International Knee Documentation Committee score

## Abstract

Little attention has been paid to knee muscle strength after ACL rupture and its effect on prognostic outcomes and treatment decisions. We studied hamstrings (H) and quadriceps (Q) strength correlation with a patient-reported outcome measures score (International Knee Documentation Committee, IKDC), anterior tibial translation (ATT), and time post-injury in 194 anterior cruciate ligament deficient patients (ACLD) who required surgery after a failed rehabilitation program (non-copers). The correlation between knee muscle strength and ATT was also studied in 53 non-injured controls. ACLD patients showed decreased knee muscle strength of both the injured and non-injured limbs. The median (interquartile range) values of the H/Q ratio were 0.61 (0.52–0.81) for patients’ injured side and 0.65 (0.57–0.8) for the non-injured side (*p* = 0.010). The median H/Q ratio for the controls was 0.52 (0.45–0.66) on both knees (*p* < 0.001, compared with the non-injured side of patients). The H/Q, ATT, and time post-injury were not significantly correlated with the IKDC score. ATT was significantly correlated with the H/Q of the injured and non-injured knees of patients, but not in the knees of the controls. Quadriceps strength and H/Q ratio were significantly correlated with ATT for both limbs of the patients. IKDC score correlated significantly with the quadriceps and hamstrings strengths of the injured limb but not with the H/Q ratio, ATT or time passed after injury.

## 1. Introduction

Non-operative treatment of anterior cruciate ligament (ACL) rupture is a better option over surgical reconstruction for many surgeons. Patients who return to their pre-injury activities after ACL rupture, following non-operative treatment without ACL reconstruction, are defined as “copers”, whereas individuals who require surgical intervention to resume their pre-injury activities are defined as “non-copers” [1].

Fitzgerald et al. [2] developed a test battery to identify potential copers after an ACL rupture, concluding that they are patients who demonstrate good self-reported activity performance and knee function. Non-copers are those with instability and/or poor self-reported or performance-based knee function. However, after an adequate neuromuscular and strength-training program, a significant percentage of patients initially classified as non-copers can become copers, avoid anterior cruciate ligament reconstruction (ACLR), and successfully perform sport activities [3]. These authors observed that persistent non-copers fared poorly after 2 years and advised a more intensive preoperative rehabilitation program. When treated with physical therapy alone, 70% of the patients classified as non-copers early after injury became true copers after 1 year [3].

Becker and Karlson [4] indicated that more attention should be paid to muscle strength and knee function instead of surgical techniques for ACLR. ACL rupture elicits changes in the extensor and flexor muscles of the knee. There is a deficit in the activation of the quadriceps bilaterally [5] and a smaller volume of the leg muscles of patients with ACL injuries compared with those of healthy controls [6]. The H/Q ratio is the balance between the flexor and extensor muscles of the knee. Therefore, it should be related to the subjective functional capacity of the knee, which is a better outcome measure over an isolated measurement of extensor or flexor strength [7,8]. The H/Q ratio is a determining factor of knee function to be considered in the progress of rehabilitation and return to sports after an ACL injury [9]. It is altered in an ACL-deficient knee compared with that in the uninjured contralateral limb [8,10]. In non-copers, identifying the values of knee muscle strength before the operation could help to address the rehabilitation program to improve conservative treatment outcomes.

ACL injury can elicit excessive ATT and instability of the joint with the possibility of early arthrosis [11,12,13,14]. A difference in ATT of >3 mm between the injured and non-injured knees is considered pathological. However, it remains unclear whether ATT affects patient-reported outcome measures (PROMs), and little is known about the association between patient-related outcome scores and time post-injury.

Knee muscle strengthening is an essential part of the rehabilitation program for conservative and operative treatment of ACL rupture. PROM and ATT, among others, are crucial parameters that help to decide on conservative or surgical treatment. Knowing if there is a significant correlation between knee strength and these parameters in a population of ACL deficient patients who chose to be operated on after failing conservative treatment (non-copers) would help implement changes in the rehabilitation program to improve the results of the non-operative treatment. It was hypothesized that knee muscle strength correlates negatively with IKDC scoring and ATT in non-copers ACL deficient patients. The second hypothesis was that IKDC does not significantly correlate with ATT and time passed after injury.

## 2. Materials and Methods

### 2.1. Ethics Statements

This study was approved by the Local Human Research Ethics Committee (protocol number CEIH-2017-11) and performed in accordance with the 1964 Helsinki Declaration. Written informed consent was obtained from all participants prior to their participation in the study.

### 2.2. Design

This cross-sectional observational study recruited 194 patients presenting with knee instability after a unilateral rupture of the ACL and 53 non-injured controls. Patients answered subjective questionnaires and underwent objective tests within 24 h prior to an arthroscopic ACL reconstruction. The same objective tests were performed for the controls.

### 2.3. Participants

The participants were selected from a population of patients scheduled for ACLR at the same hospital between January 2016 and December 2018. All patients were diagnosed with symptomatic rupture of the ACL with positive Lachman and Pivot shift test results. The preoperative diagnosis was complemented by nuclear magnetic resonance findings for all but four patients. Diagnostic certainty was confirmed through arthroscopy. Controls were selected from a population without leg injury, whose age and sex were similar to that of the patients; they were recruited from users of a fitness center located close to the hospital.

The inclusion criteria for the patients were as follows: patients who consented to study investigations; those who presented with symptoms and signs of instability and were scheduled for ACLR; those with unilateral ACL rupture >80% (according to the arthroscopic view) caused by a knee injury >4 weeks prior to ACLR; and those whose range of movement of the injured knee was >90% of that of the non-injured knee. The inclusion criteria for the controls were as follows: individuals without injury in both legs, individuals without physical restrictions in activities of daily living, and individuals who agreed to participate in the study.

The exclusion criteria for the patients were as follows: patients with symptoms and/or signs of insufficiency of any of the knee ligaments other than ACL and history of ligament rupture or tendon injury of the injured or uninjured knee and femur or tibial fracture. Patients with a chondral or meniscal lesion were not excluded.

### 2.4. Measures

Self-reported knee function was evaluated using the International Knee Documentation Committee (IKDC) form. The objective test was performed on both knees of the patients (first on the injured knee, and subsequently on the uninjured knee) and controls by the same researchers.

ATT was assessed using the KT-1000 arthrometer (MEDmetric, San Diego, CA, USA). The isometric strength of the quadriceps and hamstrings of both knees was measured using a hand-held dynamometer [15,16] (HHD) (MicroFET3; Hoggan Health Industries, West Jordan, UT, USA). Maximal force was expressed in Newton (N).

Participants were taught to perform isometric contractions of the knee muscles. They performed warm-up exercises for 5 min and 2 practice trials of the tests, rested for 30 s, and subsequently performed the 3 measurement trials. Extension strength was measured according to the protocol described in previous studies [15,16]. The HHD was positioned 2 cm proximal to the lateral malleolus. Isometric quadriceps strength was tested with the hips and knees in 90° and 60° of flexion, respectively. Hamstring isometric strength was measured with the participant positioned in the prone position with an arm crossed under the forehead on a stretcher with the knees at 3° flexion and hips at 0° flexion. The HHD was placed on the calcaneus, at the level of the Achilles tendon insertion. In all strength tests, the dynamometer was secured to the leg with an immovable strap.

Measurements were performed three times for each limb, and the average was used for statistical analysis. If the difference between one of the three intratest data and the others was >10%, the measurement was retaken. During the tests, participants were encouraged to maximally contract the muscle.

### 2.5. Statistical Analyses

The normality of age in cases and controls was tested using the Kolmogorov–Smirnov test. The results are summarized as mean and standard deviation (SD) in each group. The means were compared using the Student’s *t*-test. Gender is summarized as frequencies and percentages and compared using the chi-square test (𝜒^2^). For within-subject analysis (comparison of knees), results were expressed as medians and interquartile ranges (IQR = 25th–75th percentile) and were compared using the Wilcoxon test for dependent samples. For between-group analysis, markers were summarized as means (95% CI) adjusted for age and gender using least squares. Concordances between continuous markers were determined by the intraclass correlation coefficient (ICC), which was estimated based on 95% confidence intervals (CIs). All linear correlations were evaluated by the Spearman correlation coefficient (R). A power analysis was performed for the Spearman’s correlations. The powers were estimated for the observed value of the correlation and the sample size. The values of the markers and scales obtained throughout the follow-up period were adjusted for LOESS (Local Regression). For the regression contrasts, the *p*-values were obtained using the likelihood ratio test. When appropriate, a power analysis result was represented using the bootstrap. Statistical significance was set at *p* < 0.05. Data were analyzed using the R statistical package, version 3.6.1 (R Development Core Team, 2019. R Foundation for Statistical Computing, Vienna, Austria)

## 3. Results

Of 194 patients, 77 (39.69%) had no meniscal rupture, 40 (20.61%) had external meniscus rupture, 51 (26.28%) had internal meniscus rupture, and 26 (13.4%) had rupture of both menisci. There were 21 patients (10.82%) who showed a type of minor chondral lesion.

Table 1 summarizes the demographic characteristics of the patients and controls.

Table 2 shows the comparisons of the quadriceps and hamstrings strengths, H/Q ratio, and ATT between the patients and controls. Both quadriceps and hamstrings strength of patients were significantly greater on the non-injured side than on the injured side (*p* < 0.001); the H/Q ratio was slightly greater on the non-injured side than on the injured side (*p* = 0.010); and ATT was significantly greater on the injured side than on the non-injured side (6 versus 2.5 mm; *p* < 0.001).

Although differences were less evident, both quadriceps and hamstring strength was greater on the dominant side than on the non-dominant side of the controls (*p* ≤ 0.001). However, neither the H/Q ratio nor knee displacement showed significant differences between both limbs in the control group. Quadriceps strength of the non-injured side of the patients was significantly less than the average of the quadriceps strength of both limbs in the controls (*p* < 0.001). The H/Q ratio of the non-injured limb was significantly greater than the average of the H/Q ratio of both limbs in the controls (*p* < 0.001). ATT was not significantly different between the non-injured side of patients and the average of both limbs in the controls.

Figure 1 shows the simultaneous scatter plot of the H/Q ratio for the two sides; namely, for the patients (injured versus non-injured side) and for the controls (dominant versus non-dominant side). For the controls, the estimated ICC was 0.932 (95% CI = 0.886–0.969), indicating strong agreement between the values of both sides. However, this agreement was lower between the injured and non-injured sides in the patients (ICC = 0.834; 95% CI = 0.786–0.872).

Quadriceps strength was significantly correlated with ATT for both injured and non-injured limbs of the patients (Spearman R = 0.214, *p* = 0.003 versus (vs.) 0.18, *p* = 0.012; Figure 2).

Hamstrings strength was significantly correlated (negatively) with ATT on the non-injured side (Spearman R = −0.157, *p* = 0.029) but not on the injured side (Spearman R = −0.031, *p* = 0.664; Figure 2). The H/Q ratio was significantly (negatively) correlated with ATT on the injured and non-injured sides (Spearman R = −0.153, *p* = 0.034 vs. −0.291, *p* < 0.001; Figure 2).

Quadriceps and hamstrings strength of the injured side was significantly correlated with the IKDC score (Spearman R = 0.201, *p* = 0.005 vs. 0.181, *p* = 0.011; Figure 3).

However, neither the H/Q ratio (Spearman R = 0.008, *p* = 0.915; Figure 3) nor tibial displacement (Spearman R = −0.025, *p* = 0.728; Figure 3) was significantly correlated with the IKDC score. No significant correlation was observed between the IKDC score and quadriceps or hamstrings strength, H/Q ratio, or ATT on the non-injured side (Figure 4).

Hamstrings strength on the injured side was nearly significantly correlated with time post-injury (Spearman R = 0.158, *p* = 0.052). No significant correlation was observed between time post-injury and quadriceps strength, the H/Q ratio, IKDC, or ATT (Figure 5).

## 4. Discussion

In this study, quadriceps and hamstrings strengths and the H/Q ratio were significantly higher on the non-injured side than on the injured side of patients; the H/Q ratio was significantly higher and quadriceps strength was significantly lower on the non-injured side of patients compared with the average H/Q ratio and quadriceps strength on both legs of the controls. This finding has also been observed by other authors [5,6,10,17] and justified by factors such as bilateral arthrogenic muscle inhibition and atrophy due to a decrease in physical activity after injury.

The H/Q ratio can be used as an indicator of muscular imbalance around the knee joint, and it is suggested to be more important than the maximal torque in the assessment of muscle function [7,8]. The H/Q values of both legs of patients and controls in our study were within the range of 0.5 to 0.75, which are considered normal [10,18,19]. However, the average H/Q ratio was higher in both legs of the patients than in those of the controls, suggesting a bilateral decrease in quadriceps activity in patients with ACLD [5,10,17,20,21]. In a meta-analysis, Kim et al. [8] found that the reduction in quadriceps muscle strength was about three times greater than the reduction in hamstrings muscle strength. These uneven reductions in thigh muscle strength contributed to muscle imbalance, as shown by the higher H/Q ratio of the injured than of the uninjured limbs. Since muscle strength of both legs is affected after ACL rupture, using the uninjured leg as the control leg for muscle strength comparisons after ACL injuries may produce false results [22].

The IKDC results of most of our patients were less than 80. A PROM such as the IKDC may serve as a valuable screening tool for identifying quadriceps strength deficits in patients with ACLD [23] and the possibility of a successful return to sports [24]. The IKDC results of our patients were significantly correlated with quadriceps and hamstrings strength of their injured knee, but not with those of the uninjured one. Despite being significant, correlations between quadriceps strength and IKDC score and ATT were weak. Other variables not considered could influence these results. Adjusting the power by age, sex or normalized body mass index might change the correlations observed. Although there are huge references about the importance of quadriceps strength for a successful recovery after ACLR, little is known about the correlation of muscle strength and PROMs before the operation and how this relationship could influence treatment decisions. Our results differ from others that did not find significant correlations between IKDC score and muscle strength in ACLD patients [17]. As in our study, Hohmann et al. [10] observed no significant relationship between the H/Q ratio and several PROM results in non-operated ACLD patients. However, the correlation was significant after ACLR.

PROM questionnaire scores are significantly lower in patients with ACLD than in healthy individuals. Although questionnaire scores show better results after ACLR, patients rarely achieve the same score observed in healthy populations [25]. Previous studies reported that in a group of participants with ACL injury capable of performing hop and side cut tasks, there was a significantly lower subjective function in all PROMs and knee extensor torque [25].

One important finding of the present study was that time post-injury did not significantly correlate with the hamstring strength, the quadriceps strength, H/Q ratio, IKDC, and ATT in patients with ACLD, confirming our hypothesis. A recent systematic review by Keizer and Otten [26] found that ATT was significantly higher in chronic than acute ACL injuries. Nguyen et al. [27] observed that before reconstruction (>6 months), the chronic group achieved higher baseline IKDC scores than the acute group (<3 months). They concluded that the chronic group participated less in pivoting and cutting sports but had improved pain/function. Interestingly, Cristiani et al. [24] found that a time from injury to surgery longer than 3 months significantly reduced the odds of achieving symmetrical hamstring muscle strength 6 months after ACLR. The time elapsed since an injury can possibly reduce physical activity. Perhaps preoperative rehabilitation prioritizes muscular strengthening of the quadriceps. Both situations can cause hamstring deconditioning. In these studies, patients were divided into two groups (acute and chronic) empirically. To the best of our knowledge, our study is the first to correlate real-time post-injury with objective and subjective parameters of ACL-deficient knees.

### 4.1. Study Limitations

This study has several limitations. First, strength was not normalized to body mass or weight as recommended in previous studies [16]. Similar to previous studies [7,8], we considered that the H/Q ratio rather than quadriceps or hamstrings strength is a better determinant of knee muscle function. The H/Q ratio is a quotient, and its value does not change when using absolute values or values normalized to body mass. We used an HDD to measure force, since there is currently sufficient evidence that measuring isometric strength using manual devices provides enough reliability and reproducibility, and HDD is a very useful tool in normal clinical settings [15,16,28,29,30,31]. Several studies have confirmed that H/Q ratio values are highly dependent on the knee joint angle at which the measurement is taken [8,19,32]. We do not know if our results could be different with tests carried out at other knee flexion angles. Another concern could be that we did not consider the length of the lever arm used to measure muscle strength in our cases [33].

Second, we used only the IKDC questionnaire as a PROM, which is the most used PROM [25,34,35]. This study could not assess if the muscle strength of our patients significantly correlates with other PROM results. Third, no specific rehabilitation protocol was prescribed prior to surgery because the patients were recruited from different health facilities. We used the non-injured leg as a “healthy” comparison for the injured leg, despite a previous report that the non-injured leg is also affected by the decrease in strength [22]. Based on our results that showed significant differences between the healthy knees of patients and that of controls, the effect of ACL injury on the strength of the non-injured knee can be possibly determined if the pre-injury values were obtained. Therefore, it is more appropriate to use normative values of knee muscle strength, adjusted for sex, weight, and activity level [22]. Another limitation is that our patient population was patients with ACLD who opted for surgery due to the persistence of instability after several weeks of rehabilitation (non-copers). Therefore, the findings of this study could not be compared with non-copers who did not opt for surgery, and we could not determine the number of non-copers who could became copers after a longer rehabilitation program.

Finally, correlations drawn from cross-sectional studies cannot establish the temporal relationship that links cause with effect. In observational studies, the relationship between the variables is vulnerable to bias from those unobserved variables that were not measured. These variables might be affecting the study variables, influencing the interpretation of the data. Thus, interpretation of correlational findings must be cautious until further research is completed [36]. In our study, a significant, although weak, correlation was observed between IKDC scoring and muscle strength of the injured knee, but not the uninjured one. From these results it cannot be assumed that improving the knee strength will increase the IKDC results in ACLD patients. However, they support the finding observed by Thoma et al. [3] that progressive neuromuscular and strength training can change the status of ACLD patients from non-copers to copers, avoiding operations in a substantial number of cases. The correlations observed in our study advise us to carry out the next stage of clinical research through randomized controlled clinical trials.

### 4.2. Clinical Implications

In this study, the quadriceps strength was weaker than the hamstring strength in patients’ injured and uninjured knees, independently of the time passed after the injury. Muscle strength of the injured knees significantly correlated with the IKDC scoring (positive correlation) and the ATT, two crucial parameters for ACLD treatment decisions. Tests were carried out within 48 h before patients and doctors decided on ACLR due to symptomatic instability of the knee after a minimum of 4 weeks of rehabilitation. There was no control about the rehabilitation program followed by the patients before the operation and how knee muscle strengthening was carried out. Perhaps with more prolonged time and specific muscle training, some of our patients would not have needed an ACLR. Logersted et al. [37] found that deficits in pre-operative quadriceps strength influence self-reported function 6 months after surgery. Pre-operative quadriceps and hamstring muscle strength deficits may significantly negatively impact functional performance following ACLR [38].

Thoma et al. [3] observed that patients classified initially as non-copers can become copers after an adequate neuromuscular and strength conditioning program [3]. The findings of our study advise strengthening quadriceps and hamstring of both the injured and uninjured knees after ACL rupture, checking the knee strength objectively regularly to implement changes in the program if necessary [38].

## 5. Conclusions

Patients with ACLD (non-copers) showed a decrease in knee muscle strength of both the injured and non-injured limbs. Quadriceps strength and H/Q ratio were significantly correlated with ATT for both the injured and non-injured limbs of the patients. The IKDC score correlated significantly with quadriceps and hamstrings strength of the injured limb but not with the H/Q ratio, ATT or time passed after injury. After an ACL rupture, muscle strength conditioning of both the injured and non-injured limb should be considered and implemented as soon as possible.

## Figures and Tables

**Figure 1 ijerph-18-13303-f001:**
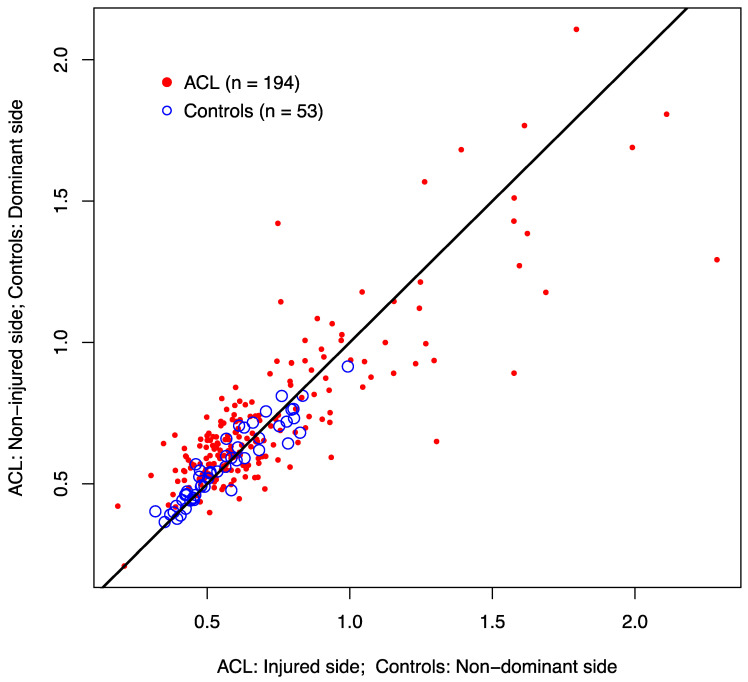
Agreement between the H/Q ratio of the knees. ACL: healthy versus affected (ICC = 0.872; 95%CI = 0.786–0.872). Control subjects: right versus left sides (ICC = 0.932; 95% CI = 0.886–0.960); H/Q, hamstring/quadriceps; ACL, anterior cruciate ligament; ICC, intraclass correlation coefficient; CI, confidence interval.

**Figure 2 ijerph-18-13303-f002:**
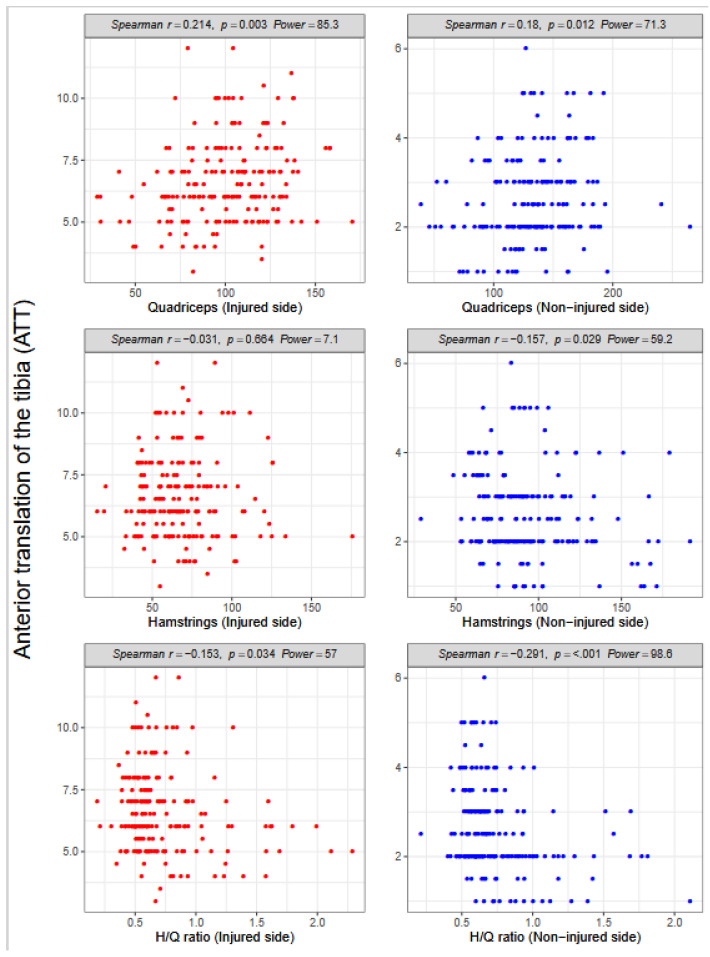
Anterior tibial translation (ATT) according to strength in the quadriceps and hamstrings and side (injured and non-injured); H/Q, hamstring/quadriceps.

**Figure 3 ijerph-18-13303-f003:**
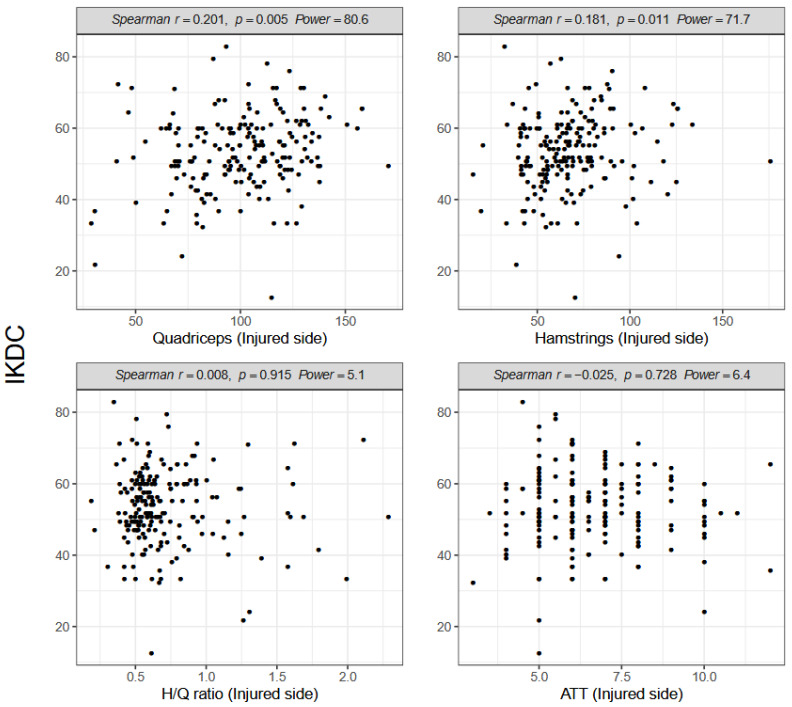
Spearman correlations between the International Knee Documentation Committee (IKDC) score, muscle strength and anterior tibial translation (ATT) on the injured side; H/Q, hamstring/quadriceps.

**Figure 4 ijerph-18-13303-f004:**
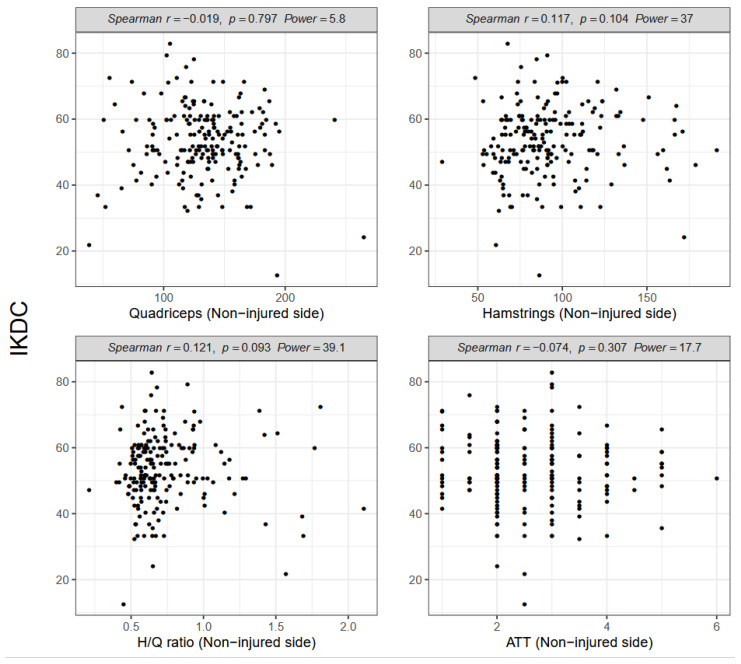
Spearman correlations between the International Knee Documentation Committee (IKDC) score, muscle strength and anterior tibial translation (ATT) on the non-injured side; H/Q, hamstring/quadriceps.

**Figure 5 ijerph-18-13303-f005:**
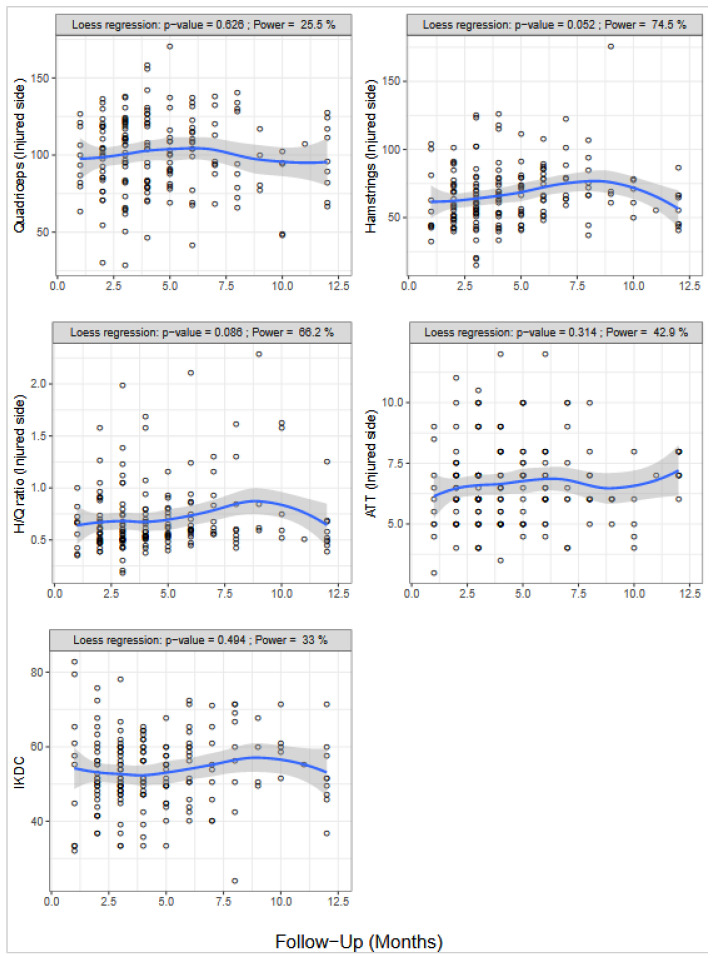
Muscle strength, the hamstring/quadriceps (H/Q) ratio, and International Knee Documentation Committee (IKDC) score according to the time post-injury (months). Data smoothing was performed using the LOESS function.

**Table 1 ijerph-18-13303-t001:** Demographic data of the patients and controls.

	ACL N = 194	Controls N = 53	*p*
Age	29.5 ± 9.9	29.9 ± 8.9	0.814
Sex (Male)	155 (79.9%)	46 (86.8%)	0.253
IKDC scoring	53.3 ± 10.7	-	-
Time from injury in months	4 (3–7)	-	-

Data are means ± SD and median (25–75% IQR).

**Table 2 ijerph-18-13303-t002:** Muscular strength (Newtons) and anterior translation of the tibia (ATT).

	ACL (N = 194)			Controls (N = 53)				
	Injured Side	Non-Injured Side	*p* *	Non-Dominant Side	Dominant Side	*p* *	Average of Both Sides	*p* **
Q	101.9 (82.4–119)	133.4 (116–154.4)	<0.001	166.4 (145.1–190.5)	177.6 (157.4–196.5)	<0.001	174.8 (152.6–192.6)	<0.001
H	65.5 (52.17–78)	86.6 (73.6–103.9)	<0.001	84.1 (74.9–108.9)	90.2 (77.4–115.3)	<0.001	89.3 (76.5–111.5)	0.487
H/Q ratio	0.61 (0.52–0.81)	0.65 (0.57–0.8)	0.010	0.51 (0.44–0.66)	0.54 (0.46–0.66)	0.292	0.52 (0.45–0.66)	<0.001
ATT (mm)	6 (5.1–7.5)	2.5 (2–3)	<0.001	2 (2–3)	2 (2–3)	0.575	2.25 (2–2.75)	0.454

Data are medians (25–75% IQR). Q: quadriceps. H: hamstrings. ATT: anterior tibial translation. (*) Wilcoxon test for paired data. (**) Unpaired Wilcoxon test for comparing the non-injured side (ACL) with the average of both sides (controls).

## Data Availability

The data that support the findings of this study are available from the corresponding author, G.L.G., upon reasonable request.

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
