# Peer review of "Correlation among Knee Muscle Strength and Self-Reported Outcomes Score, Anterior Tibial Displacement, and Time Post-Injury in Non-Coper Anterior Cruciate Ligament Deficient Patients: A Cross-Sectional Study"

_ijerph, 2021, doi:10.3390/ijerph182413303_

Round 1
Reviewer 1 Report
This paper shows the correlation between knee muscle strength and self-reported outcomes score, anterior tibial displacement, and time post-injury. The author emphasised the significance of correlation but did not address the strength of correlation. Most of the correlations shown in this paper are weak so the contribution and impact of this study are not clear. This paper should be revised properly to highlight its rationale, application to clarify its contribution and impact. Further detail such as the typical ratio between coper and non-coper should be provided in the introduction. Table 2 should be correct to show properly in the context.

Author Response
Thank you for your comments, which have been very useful. Please see the attachment with our answers

Reviewer 2 Report
Congratulations, your paper and your research is very interesting and has a high quality.
However, you should take care of the layout as there are some tables that do not fit completely in the PDF (e.g. table 2) and there are large blank spaces (such as after table 1).
Besides that there are some other minor flaws. For example, the acronym PROM (I understand it refers to "Patient Reported Outcome Measures") that first appears on line 83, has not been previously defined. Also, LOESS (which appears on line 180, among others), should be capitalized because it is another acronym.
And finally, the page numbering starts in the middle of the article instead of at the beginning.
Author Response
Thank you for your comments, which have been very useful. Please see the following our answers
Congratulations, your paper and your research is very interesting and has a high quality.
However, you should take care of the layout as there are some tables that do not fit completely in the PDF (e.g. table 2) and there are large blank spaces (such as after table 1). This has been already corrected
Besides that there are some other minor flaws. For example, the acronym PROM (I understand it refers to "Patient Reported Outcome Measures") that first appears on line 83, has not been previously defined. Also, LOESS (which appears on line 180, among others), should be capitalized because it is another acronym. These have been already corrected
And finally, the page numbering starts in the middle of the article instead of at the beginning. This has been already corrected
Reviewer 3 Report
The authors showed a significant correlation of quadriceps strength and H/Q ratio with ATT for patients of both limbs, which is relevant for the clinical outcomes.
The article is well-written and well-organized.
Minor points:
-Even though the authors mentioned why the strength was not normalized to body mass or weight, I would suggest the authors to add the normalized data in the article for comparison.
Author Response
Thank you for your comments, which have been very useful. Please see following our answers:
The authors showed a significant correlation of quadriceps strength and H/Q ratio with ATT for patients of both limbs, which is relevant for the clinical outcomes.
The article is well-written and well-organized.
Minor points:
-Even though the authors mentioned why the strength was not normalized to body mass or weight, I would suggest the authors to add the normalized data in the article for comparison. Unfortunately, this is a retrospective study and we do not have the data of weight and height for all the patients. This aspect was pointed out as one limitation of the study.
Reviewer 4 Report
This study suggested interesting results and experimental suggestions.
Also, this paper is well written with logical flow.
However, I found some minor limitations and suggest revision related these issues.
This paper deserves to be published after minor revisions.
1. Check the correct reference format according to the author guidelines of this journal.
2. Check for format consistency in each cell of the table, and check for missing abbreviations under the table.
Author Response
Thank you for your comments, which have been very useful.
- Check the correct reference format according to the author guidelines of this journal. It has been already corrected
- Check for format consistency in each cell of the table, and check for missing abbreviations under the table. It has been already corrected
Round 2
Reviewer 1 Report
Thanks for the authors spending time to revise the paper. However, the authors did not address the issue of weak correlations. A significant correlation can be a weak or a strong correlation. The results in this paper show significant weak correlations between knee strength and the selected parameters, so they could not help implement changes in the rehabilitation program to improve the results of the non-operative treatment. The discussion of weak correlation is required in the next revision, and further revisions are also needed for the relevant arguments and statements.
Author Response
Thanks for your suggestions to our reviewed manuscript. We have added some comments to the discussion following these suggestions. We understand that we found a weak correlation between muscle strength with IKDC and ATT. We have outlined this in the new version of the manuscript, and we have added some comments in the limitations section. Despite this low correlation found, we consider that our results are in line with other previous references, allowing us to recommend muscle-strengthening before deciding about ACLR. We agree that further research about the exact contribution of knee muscle strength after ACL rupture is mandatory to advise conservative treatment rather than ACLR.